# Evaluation of Minimum Unit Pricing of Alcohol: A Mixed Method Natural Experiment in Scotland

**DOI:** 10.3390/ijerph17103394

**Published:** 2020-05-13

**Authors:** Clare Beeston, Mark Robinson, Lucie Giles, Elinor Dickie, Jane Ford, Megan MacPherson, Rachel McAdams, Ruth Mellor, Deborah Shipton, Neil Craig

**Affiliations:** 1Place and Wellbeing Directorate Public Health Scotland, Glasgow G2 6QP, UK; Lucie.giles@nhs.net (L.G.); elinor.dickie@nhs.net (E.D.); Jane.ford3@nhs.net (J.F.); Rachel.mcadams@nhs.net (R.M.); Deborah.Shipton@nhs.net (D.S.); neil.craig@nhs.net (N.C.); 2Institute for Social Science Research, University of Queensland, Brisbane, QLD 4068, Australia; mark.robinson@uq.edu.au; 3Public Health, NHS Grampian, Aberdeen AB15 6RE, UK; Megan.macpherson@nhs.net; 4Public Health, NHS Lanarkshire, Glasgow G71 8BB, UK; ruth.mellor@nhs.net

**Keywords:** alcohol, price, Scotland, evaluation, minimum unit pricing, natural experiment

## Abstract

In May 2018, Scotland became the first country in the world to implement minimum unit pricing (MUP) for all alcoholic drinks sold in licensed premises in Scotland. The use of a Sunset Clause in the MUP legislation was a factor in successfully resisting legal challenges by indicating that the final decision on a novel policy would depend on its impact. An overarching evaluation has been designed and the results will provide important evidence to inform the parliamentary vote on the future of MUP in Scotland. The evaluation uses a mixed methods portfolio of in-house, commissioned, and separately funded studies to assess the impact of MUP across multiple intended and unintended outcomes related to compliance, the alcoholic drinks industry, consumption, and health and social harms. Quantitative studies to measure impact use a suitable control where feasible. Qualitative studies assess impact and provide an understanding of the lived experience and mechanism of change for key sub-groups. As well as providing important evidence to inform the parliamentary vote, adding to the international evidence on impact and experience of alcohol pricing policy across a broad range of outcomes, this approach to evaluating novel policy interventions may provide guidance for future policy innovations.

## 1. Introduction

Following a period of high and increasing rates of alcohol-related harm in Scotland [1], in 2009 the Scottish Government proposed a comprehensive strategy that aimed to reduce population levels of alcohol consumption and, in turn, levels of associated health and social harms [2]. The introduction of minimum unit pricing (MUP) for alcohol was a central part of this strategic approach and, following an initial unsuccessful attempt, the necessary primary legislation was passed by the Scottish Parliament in 2012. A legal challenge followed, ending when the UK Supreme Court ruled in favour of the Scottish Government in late 2017. The initial level was set at 50 pence per unit (ppu), estimated to affect 47% of all alcohol sold through the off-trade in Scotland in 2017 [3]. The policy was implemented on 1 May, 2018, when Scotland became the first country to set a strength-based floor price which applied to all alcohol sold in or through licensed premises.

### 1.1. Existing Evidence

The rationale for MUP is based on strong and consistent systematic review evidence on the relationship between alcohol price, consumption, and related-harm. Alcohol consumption is associated with a substantially increased risk of all-cause mortality and, while the size of the effect may vary, there is evidence from a number of studies that increases in price result in a decrease in consumption and associated harms [4,5]. There is some evidence that pricing interventions may in particular target young people (aged under 18 years), young adult ‘binge’ drinkers, and heavy drinkers [4].

Estimates of alcohol price elasticity suggest that a 10% rise in price might be expected to reduce demand for alcohol by about 5% [6,7]. Price elasticity varies by levels of consumption, with drinkers becoming proportionally less responsive to price as consumption increases [8], although in absolute terms the reduction in units of alcohol consumed is greatest in heavier drinkers [9].

Using existing evidence on the relationship between alcohol price, consumption, and harm, in 2016 it was estimated that a 50 ppu minimum unit price would reduce population alcohol consumption in Scotland by 3.5% per year, leading to over 120 fewer alcohol-attributable deaths and over 2000 fewer alcohol-attributable hospital admissions per year when the policy reached its full effect (at 20 years), with effects most pronounced among those drinking at harmful levels, particularly those on lower incomes [10].

Empirical evidence on the impacts of minimum pricing for alcohol is available from Canada, where a form of minimum pricing applies in each of the 10 provinces. There is variation in the extent and frequency to which different drink types and outlets are affected by different forms of minimum pricing in these provinces [11]. Retrospective analysis using population level administrative data from British Columbia has shown that as minimum alcohol prices increase, there is an associated decrease in population consumption [12], hospital admissions [13], deaths [14], and crimes [15] related to alcohol. Similar results have been observed in Saskatchewan, which has the form of minimum pricing most similar to Scotland. Here, a 10% increase in minimum price reduced consumption of all alcohol by 8% [16].

A limited form of MUP was introduced in the Russian Federation in 2010 as part of a comprehensive set of alcohol control policies introduced at various points from 2000 onwards. Efforts initially focused on restricting the production of alcohol, with increases in alcohol excise taxes, substantial reductions in the availability of alcohol and MUP for vodka and other spirits introduced in 2011. Since 2004, there has been a fluctuating downward trend in both alcohol consumption and related mortality, with a slowdown in the decline between 2014 and 2018, when there was a loosening of some policy measures, including a decrease in the MUP [17].

The impact in population subgroups is likely to vary. Heavier drinkers are more likely to mitigate the effects of price increases by consuming lower quality products, although the setting of a floor price may limit the opportunities for ‘trading down’ in this way [8]. A study with homeless drinkers found the most common strategies employed when alcohol became less affordable (due to changes in individual income or price) included re-budgeting (e.g., foregoing essentials such as food so as to afford alcohol), waiting for money, and going without alcohol. Other consequences reported by at least a third of the sample included illicit drug use, drinking non-beverage alcohol, stealing alcohol, and seeking help or treatment [18]. However, in a study of dependent drinkers in Edinburgh during a period when alcohol became less affordable, there was little evidence of acquisitive crime or substituting other harmful substances (such as non-beverage alcohol or illicit drugs) for alcohol. However, it was noted that the study took place before the introduction of MUP and participants reported switching to cheaper products still available at the time [19].

Alcohol-related harms have shown to be socially patterned, disproportionately affecting the most disadvantaged at a given level of alcohol consumption [20]. As such, the introduction of minimum pricing may reduce inequalities in alcohol-related harms.

### 1.2. The Purpose of This Evaluation

MUP in the form it takes in Scotland has not been implemented elsewhere and it is important to add to the evidence base on the impact and experience on alcohol pricing policies across a range of outcomes. Furthermore, the Scottish legislation includes a Sunset Clause requiring a positive parliamentary vote if MP is to continue beyond an initial six year period, and a Review Clause requiring that the Scottish Parliament must be provided with a report on the operation of MUP and its impacts before this vote [21]. The review report is to cover the impact on:Alcoholic drink producers and licence holders;The five licensing objectives (covering crime and disorder, public safety, public nuisance, public health, and protecting children and young persons from harm);Different groups (defined by age, gender, social and economic deprivation, and levels of alcohol consumption), where possible.

In their ruling, the Supreme Court recognised the experimental nature of MUP and judged the inclusion of the Sunset and Review Clauses to be important in reaching their decision [22].

The Scottish Government has tasked NHS Health Scotland (now part of Public Health Scotland) with delivering an evaluation of MUP that will form the basis of the review report. The aim is to provide robust and comprehensive evidence on the impact of MUP in Scotland for to generate knowledge and inform the Parliamentary vote on the future of MUP.

## 2. Materials and Methods

### 2.1. Evaluation Design

The evaluation takes a theory-based approach to answer the evaluation questions:To what extent has MUP contributed to reducing alcohol-related health and social harms in Scotland?Are some people and businesses affected (positively and negatively) more than others?

Having a clear theoretical understanding of how a social or public health policy intervention is expected to achieve its effect is useful in a natural experiment design when it is difficult or impossible to use other experimental methods to control exposure [23] and where there are many potential interacting outcomes across a range of domains [24].

A theory of change (Figure 1) that shows the main chain of outcomes that may result from MUP implementation has been developed, taking account of the existing evidence, legislative requirements, and stakeholder concerns.

The theory of change shows the main expected chain of outcomes whereby compliance with MUP results in an increase in the price of low cost, high strength alcohol, reducing alcohol purchasing and consumption and in turn reducing alcohol-related health and social harms. MUP may stimulate the alcoholic drinks industry to make changes in product range, alcoholic strength, pack size, price of alcohol already >50 ppu, and/or marketing strategies. MUP may also lead to changes in attitudes to MUP and alcohol more broadly. There may be other changes, such as substitution to non-beverage alcohol or other drugs; displacement of spending previously used for other goods or services; an increase or decrease in demand for services; and a variable economic impact on organisations that are part of the alcoholic beverage production, distribution and retail chain.

The effects of MUP will be influenced by its interaction with factors external to MUP, such as factors that influence the price of alcohol and/or disposable income, other alcohol policy, or changes in the broader social and economic determinants of health.

Finally, the effects of MUP may change over time. For example, any immediate impact of reduced availability of strong, low cost alcohol, particularly among those drinking at harmful levels, may differ from longer term effects of any change in the amount or pattern of drinking. Furthermore, the MUP in Scotland is not index linked and the real value will diminish over time.

### 2.2. Study Portfolio

For ease of communication, the theory of change has been simplified into four main outcome areas:Compliance, implementation, and attitudes—covering compliance and availability of alcohol <50 ppu. It also includes changes in social norms and attitudes to alcohol and MUP, although these may occur later in response to changes in other outcomes.Alcoholic drinks industry—covering price change, change in price distribution, reduced purchasing, product and marketing changes, and economic impact on the alcohol industry.Consumption—covering reduced individual and population consumption, and safer patterns of drinking.Health and social harms—covering reduced health and social harms, displacement of spending, substitution to non-beverage alcohol or drugs, and the impact on services.

The evaluation uses a mixed method portfolio of studies to generate robust evidence on the outcomes in the theory of change. Each outcome area will be evidenced by a number of studies from the portfolio, and many studies will contribute evidence on more than one outcome area. The portfolio includes quantitative studies designed to provide measurable quantitative estimates of change. Where robust and consistent data on exposures and outcomes are available, we will compare outcomes in Scotland with those in a control area. Where possible, statistical analysis will be undertaken to evidence the scale of change in outcomes in Scotland compared to the control area and to assess whether any differences are causally related to MUP. Other studies will quantify the extent to which outcomes have changed in Scotland after MUP even if comparison with a control area is not possible.

The portfolio includes qualitative studies to assess impact and provide an understanding of mechanisms that underpin estimates of change, how people and organisations have responded to MUP, and peoples’ lived experience of MUP in Scotland. Sub-groups of interest include those drinking at harmful levels, including those who are homeless and those under 18 years of age. This qualitative understanding is essential to improve interpretation of quantitative findings. Taken together, these will provide a robust and comprehensive assessment of the impact of MUP.

The evaluation portfolio currently consists of 12 studies funded and managed through the Monitoring and Evaluating Scotland’s Alcohol Strategy (MESAS) programme of work, and a further seven separately-funded studies. Table 1 lists these studies, with more detail in Appendix A. Protocols for the MESAS funded studies can be found on the MUP Evaluation webpages [25] and the academic literature for some of the separately funded studies [26].

### 2.3. Setting

The setting for the study is Scotland, where MUP applies, and, where possible, comparisons are made with outcomes in areas where MUP does not apply (England or England & Wales). MUP was introduced in Wales on 2 March 2020 and where appropriate the control will be adjusted accordingly. The evaluation also seeks to understand how impacts and mechanisms vary geographically within Scotland, for example, in areas close to the border with England, where possible.

### 2.4. Synthesis Across a Portfolio of Studies

Findings from across the portfolio of studies, and other appropriately critically appraised studies, will be synthesised to generate conclusions about the impact of MUP in Scotland. The effectiveness of MUP will be judged on its success in achieving the policy rationale (to reduce alcohol-related harms) via the main pathway in the theory of change (i.e., by reducing consumption, in particular in those drinking at harmful levels as a result of a price change). The other outcomes measured (e.g., economic impact on the alcoholic drinks industry) will help enhance interpretation and meet the legislative requirements. A transparent approach to synthesising findings across studies, using established and innovative methods, will be developed in preparation for the development of the final report, due in 2023. This is likely to draw on techniques including methods for synthesising qualitative evidence [27], the Decision Theory Approach [28] and Cost Consequence Analysis [29].

## 3. Discussion

MUP applies to all alcohol sold through licensed premises in Scotland. Scotland was the first country to introduce MUP and the evaluation will add to the international evidence on impact and experience of alcohol pricing policy across a broad range of outcomes. Furthermore, the use of a Sunset Clause in the MUP legislation was a factor in successfully resisting legal challenges by indicating that the final decision on a novel policy would depend on its impact. An overarching evaluation has been designed and to provide robust evidence to inform the parliamentary vote on the future of MUP in Scotland. Public Health Scotland is leading this evaluation, working with in-house, commissioned, and separately funded research teams to deliver a mixed methods portfolio of studies designed to evidence the impact of MUP on important outcomes, to understand the lived experience of key groups exposed to MUP and to explore the mechanisms by which changes occur. We will draw on systematic review of other critically appraised studies to deliver a final report that will inform the future of MUP in Scotland. This approach to evaluating novel policy interventions may provide guidance for future policy innovations.

### 3.1. Strengths and Limitations

Using a theory-based approach to design the evaluation framework strengthens causal inference in policy evaluation based on a natural experiment [23]. By collecting evidence on the expected outcomes, intermediate links in the causal chain, and on external factors that might affect the same outcomes, we can be more confident in drawing conclusions on the contribution of MUP to changes in outcomes.

To gather the evidence required, this evaluation will use the most appropriate, feasible and proportionate study designs to meet the aims of each of the component studies. For example, to isolate the specific impacts of MUP on health outcomes, we plan to use a natural experiment design, comparing health outcomes in Scotland, where MUP applies, to England or parts of England, where it does not. Such a design is based on the assumption that the main difference between the two populations is the presence or not of MUP. While another part of the UK (such as England or part of England) offers the most similar comparator population, in practice, other factors affecting outcomes, such as other alcohol policies or disposable incomes, may change in different ways in different countries at the same time as MUP. Furthermore, MUP may stimulate changes in the comparator population—for example, a MUP-driven product or price change may be rolled out in other countries of the UK. By taking a theory-based approach, with a portfolio of studies using mixed methods to gather evidence on a number of outcomes and other factors thought to influence those outcomes, in Scotland and other parts of the UK, the evaluation will help to understand whether the intended outcomes have occurred, and whether this is likely to be due to MUP rather than other confounding factors.

The MUP legislation requires that the evaluation assesses the impact of MUP across a number of domains and, where possible, across different population groups. The evaluation needs to take account of the fact that MUP is taking place within a complex system and there will be a wide range of potential outcomes (positive and negative). The whole population effects of MUP, and the effects in different sub-groups will be influenced by interactions with other elements of the system, adaptations by individuals or organisations and feedback loops. Theory-based evaluation aims to identify and measure the most important effects. Evidence from across the portfolio will be synthesised transparently and using established and innovative methods.

All studies in the portfolio are now underway. To date, deviations from protocol have been minor and limited to the compliance study, where data protection legislation and competing priorities in local authorities meant sufficient, suitable, quantitative data were not publicly available for use in the study. More recently, the current COVID-19 pandemic and the resulting social distances measures have affected delivery of some studies, including the cessation of face-to-face data collection and no access to Safe Haven facilities. Studies using administrative or market research data from 2020 and beyond are considering the best approach to dealing with the pandemic as a major factor affecting the outcomes of interest [30]. This will be a particular issue for those studies without a concurrent control and some of these have already shortened their post-MUP data period (see Appendix A). The final implications of the changes necessarily made to the evaluation will be detailed in individual study and final reports.

### 3.2. Governance

The primary purpose of this evaluation is to inform a parliamentary process for reviewing and deciding whether to continue with MUP. It is therefore important to consider both scientific and accountability processes, to ensure that the evaluation is transparent and recognised to be robust, impartial, and credible. The process and findings need to be trusted by stakeholders, including parliamentary decision makers, researchers, the public health community, service delivery partners, the alcoholic drinks industry organisations and the public. In time, it will be important to assess the impact of the evaluation and the parliamentary review on the policy process.

The development and delivery of the MESAS-funded evaluation is overseen by a multi-agency governance structure. This consists of a governance board that provides advice on study importance, the feasibility of robust design proportionate to the value of information, and the overall allocation of funding. Evaluation advisory groups provide advice to and oversee the delivery of individual or groups of MESAS-funded studies. Members of the governance groups bring research, strategic delivery, and/or contextual expertise, and there is broad representation on appropriate groups from across stakeholder groups including, but not limited to, public services, nationally commissioned organisations, the alcoholic drinks industry, Scottish Government, and academia.

## 4. Conclusions

Scotland was the first country to implement a strength-based minimum price applying to all products sold through licensed premises, and the world is watching. This important multi-component evaluation will inform the future of minimum unit pricing in Scotland. Reports from individual studies will be published as they are completed, and the final report drawing together all the studies will be published in late 2023.

## Figures and Tables

**Figure 1 ijerph-17-03394-f001:**
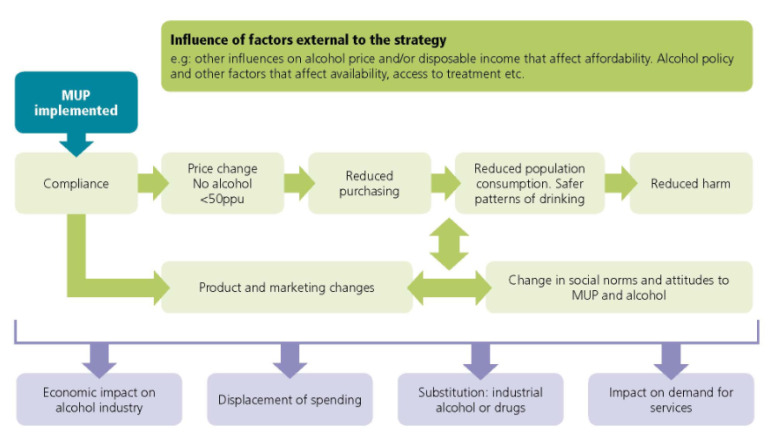
Theory of Change for minimum unit pricing for alcohol.

**Table 1 ijerph-17-03394-t001:** List of Monitoring and Evaluating Scotland’s Alcohol Strategy (MESAS) and separately funded studies, and outcome areas.

	Compliance, Implementation + Attitudes	Alcohol Industry	Consumption	Health and Social Harms
MESAS Funded Studies				
Compliance	**√**			
Economic impact		**√**		
Small retailers	√	**√**		
Price distribution	√	**√**		
Products and prices		**√**		
Sales-based consumption		√	**√**	
Harmful drinking	√		**√**	√
Children and young people: Own drinking	√		**√**	√
Health harms				**√**
Crime, public safety and public nuisance				**√**
Children + young people: Harm from others				**√**
Public Attitudes	**√**			
Separately-funded studies				
Consumption and health service impacts of MUP	√		**√**	**√**
Self-reported consumption			**√**	√
Daily survey	√		**√**	√
Homeless drinkers	√		**√**	√
Ambulance call-outs				√
Prescribing				√
Household expenditure				√

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
