# Peer review of "Evaluation of Minimum Unit Pricing of Alcohol: A Mixed Method Natural Experiment in Scotland"

_ijerph, 2020, doi:10.3390/ijerph17103394_

Round 1

Reviewer 1 Report

A very interesting and relevant paper, with implications for both the field of alcohol policy and policy evaluation. The structure of the paper made it an easy and fluid read. Also, the presentation of your theory of change as a flowchart helps the reader conceptualise your methodology.

In line 35 there is a full stop missing.

Author Response

Thank you for your comments. 

I have corrected the typo

Reviewer 2 Report

The paper provides a clear description of the multi-component evaluation that will inform the future of the minimum unit pricing in Scotland. The authors of the study might want to consider saying few words in the introduction section of the manuscript about the WHO's Alcohol policy impact case study done in Russia "The effects of alcohol control measures on mortality and life expectancy in the Russian Federation" available on the WHO's web site. In this report the minimum unit prices introduced in Russia in 2003 and later were discussed with the robust evidence suggesting their effectiveness in reducing mortality and increasing life-expectancy during the past 17 years. The continuous reduction of all-case mortality, especially among males, in Russia began right after the introduction of the MUP on vodka in 2003 which was then gradually increased during the following years.

Author Response

The paper provides a clear description of the multi-component evaluation that will inform the future of the minimum unit pricing in Scotland. The authors of the study might want to consider saying few words in the introduction section of the manuscript about the WHO's Alcohol policy impact case study done in Russia "The effects of alcohol control measures on mortality and life expectancy in the Russian Federation" available on the WHO's web site. In this report the minimum unit prices introduced in Russia in 2003 and later were discussed with the robust evidence suggesting their effectiveness in reducing mortality and increasing life-expectancy during the past 17 years. The continuous reduction of all-case mortality, especially among males, in Russia began right after the introduction of the MUP on vodka in 2003 which was then gradually increased during the following years.

Thank you for your comment and suggestion to refer to the experience in the Russian Federation the reference and the following paragraph has been added to the introduction:

A limited form of MUP was introduced in the Russian Federation in 2010 as part of a comprehensive set of alcohol control policies introduced at various points from 2000 onwards. Efforts initially focused on restricting the production of alcohol, with increases in alcohol excise taxes, substantial reductions in the availability of alcohol and MUP for vodka and other spirits introduced in 2011. Since 2004 there has been a fluctuating downward trend in both alcohol consumption and related mortality, with a slowdown in the decline between 2014-2018, when there was a loosening of some policy measures including a decrease in the MUP [17].

Reviewer 3 Report

The paper is a well written account of the design and procedures for the evaluation of the Scottish alcohol minimum pricing policy. This information is already available on the website given in the paper (p8) but the article may usefully draw the attention of a wider readership and it brings the main outline of the evaluation together in an accessible way. There are two small additions that would be useful:

  1. Are there already any publications from the evaluation? If so, perhaps they could be flagged up within the tables and references provided.
  2. The discussion could be strengthened - have there been any delays/ changes in the evaluation process and why? e.g. notes under the tables suggest this had happened. Related to this is the effects of the coronavirus. I expect the authors must be thinking about this e.g. effects on data collection and on trend data. It would be interesting to know if they have some ideas about how will they take account of this in the evaluation - it is a major unexpected extraneous factor. As the evaluation is intended to inform policy decision making in 2023, will the evaluation still be rigorous enough to properly inform the debate?

Author Response

Thank you for your comments

Point 1:

Are there already any publications from the evaluation? If so, perhaps they could be flagged up within the tables and references provided.

A good suggestion. i have added the references and links to those already published and the reference for those due to be published in the next 2 months

Point 2

The discussion could be strengthened - have there been any delays/ changes in the evaluation process and why? e.g. notes under the tables suggest this had happened. Related to this is the effects of the coronavirus. I expect the authors must be thinking about this e.g. effects on data collection and on trend data. It would be interesting to know if they have some ideas about how will they take account of this in the evaluation - it is a major unexpected extraneous factor. As the evaluation is intended to inform policy decision making in 2023, will the evaluation still be rigorous enough to properly inform the debate?

The following paragraph has been added. it is not possible to provide more detail on the implications from covid-19 at this stage as still working these through, but as you say, it will undoubtedly have implications

All studies in the portfolio are now underway. To date, deviations from protocol have been minor and limited to the Compliance study where data protection legislation and competing priorities in Local Authorities meant sufficient, suitable, quantitative data were not publicly available for use in the study. More recently, the current COVID-19 pandemic and the resulting social distances measures have affected delivery of some studies, including the cessation of face to face data collection and no access to Safe Haven facilities. Studies using administrative or market research data from 2020 and beyond are considering the best approach to dealing with the pandemic as a major factor affecting the outcomes of interest [30]. This will be a particular issue for those studies without a concurrent control and some of these have already shortened their post-MUP data period, (see Appendix A). The final implications of the changes necessarily made to the evaluation will be detailed in individual study and final reports.